# Comparison of Pregnancy Outcomes Using Different Gestational Diabetes Diagnostic Criteria and Treatment Thresholds in Multiethnic Communities between Two Tertiary Centres in Australian and New Zealand: Do They Make a Difference?

**DOI:** 10.3390/ijerph18094588

**Published:** 2021-04-26

**Authors:** Lili Yuen, Vincent W. Wong, Louise Wolmarans, David Simmons

**Affiliations:** 1School of Medicine and the Translational Health Research Institute, Western Sydney University, Campbelltown, NSW 2560, Australia; Da.Simmons@westernsydney.edu.au; 2South Western Sydney Clinical School, University of New South Wales, Liverpool, NSW 2170, Australia; Vincent.Wong1@health.nsw.gov.au; 3Diabetes and Endocrine Service, Liverpool Hospital, Liverpool, NSW 2170, Australia; 4Waikato District Health Board, Hamilton 3204, New Zealand; Louise.Wolmarans@waikatodhb.health.nz

**Keywords:** gestational diabetes, multiethnic, IADPSG, perinatal outcomes, high BMI, New Zealand, Australia, diagnostic criteria, treatment threshold

## Abstract

Introduction: Australia, but not New Zealand (NZ), has adopted the International Association of Diabetes and Pregnancy Study Groups (IADPSG) criteria to diagnose gestational diabetes (GDM). We compared pregnancy outcomes using these different diagnostic approaches. Method: Prospective data of women with GDM were collected from one NZ (NZ) and one Australian (Aus) hospital between 2007–2018. Aus screening criteria with 2-step risk-based 50 g Glucose Challenge Testing (GCT) followed by 75 g-oral glucose tolerance testing (OGTT): fasting ≥ 5.5, 2-h ≥ 8.0 mmol/L (ADIPS98) changed to a universal OGTT and fasting ≥5.1, 1-h ≥ 10, 2-h ≥ 8.5 mmol/L (IADPSG). NZ used GCT followed by OGTT with fasting ≥ 5.5, 2-h ≥ 9.0 mmol/L (NZSSD); in 2015 adopted a booking HbA1c (NZMOH). Primary outcome was a composite of macrosomia, perinatal death, preterm delivery, neonatal hypoglycaemia, and phototherapy. An Aus subset positive using NZSSD was also defined. RESULTS: The composite outcome odds ratio compared to IADPSG (1788 pregnancies) was higher for NZMOH (934 pregnancies) 2.227 (95%CI: 1.84–2.68), NZSSD (1344 pregnancies) 2.19 (1.83–2.61), and ADIPS98 (3452 pregnancies) 1.91 (1.66–2.20). Composite outcomes were similar between the Aus subset and NZ. Conclusions: The IADPSG diagnostic criteria were associated with the lowest rate of composite outcomes. Earlier NZ screening with HbA1c was not associated with a change in adverse pregnancy outcomes.

## 1. Introduction

The era of the International Association of Diabetes and Pregnancy Study Groups (IADPSG) criteria has led to significant changes in the screening strategies and diagnostic cut offs for gestational diabetes mellitus (GDM). The IADPSG recommendations published in 2010 were based on the landmark Hyperglycaemia and Adverse Pregnancy Outcomes (HAPO) study [1] which aimed to define the diagnostic cut offs where the risk of adverse pregnancy outcomes were clinically relevant. Various countries and national diabetes bodies over the last 50 years had previously adopted different diagnostic criteria, with many now converging in recent years to those recommended by the IADPSG [2]. There remain considerable differences between the screening methods and treatment targets used in New Zealand and Australia to manage GDM [3,4]. This is despite both countries’ proximity and largely publicly funded health care systems [5], with similar expenditure on health and outcomes [6].

For diagnosis of GDM, the Australian Diabetes Society in 1991 [7] recommended diagnostic criteria which involved universal screening using a 50 g GCT with a cut off of ≥ 7.8 mmol/L. The diagnostic cut off recommended for the follow up 75 g OGTT were fasting plasma glucose ≥ 5.5 mmol/L and/or 2-h ≥ 8.0 mmol/L [7]. This position was later ratified by the Australian Diabetes in Pregnancy Society (ADIPS) in a consensus statement released in 1998 [8]. In 2014, ADIPS released updated guidelines which recommended the IADPSG criteria of a universal OGTT at 24–28 weeks with fasting glucose ≥ 5.1 mmol/L, 1-h glucose ≥ 10.0 mmol/L, and/or 2-h glucose ≥ 8.5 mmol/L. There was also a corresponding lowering in the treatment targets recommended by the new guidelines. The target fasting glucose lowered from ≤5.5 mmol/L to either ≤5.1 or 5.3 mmol/L, 2-h BGL of ≤6.7 mmol/L, and the addition of a 1-h glucose ≤ 7.4 mmol/L [9]

In 1998, the New Zealand Society for the Study of Diabetes (NZSSD) adopted diagnostic criteria higher than those recommended by ADIPS to diagnose gestational diabetes [8]. The NZSSD GDM criteria include applying a universal glucose challenge test (GCT: a non-fasted 50 g glucose polymer screening test) at 24 to 28 weeks’ gestation, with a one-hour venous glucose ≥ 7.8 mmol/L leading to a follow up 75 g oral glucose tolerance test (OGTT). The 75 g OGTT diagnostic cut offs applied for GDM were a fasting glucose ≥ 5.5 mmol/L and/or a 2-h glucose ≥ 9.0 mmol/L.

In December 2014, the New Zealand Ministry of Health (NZMOH) released guidelines recommending a multi-layered screening approach where every pregnant woman should be offered a glycated haemoglobin (HbA1c) level before 20 weeks’ gestation to identify the higher risk group with undiagnosed pre-existing diabetes. Women with an HbA1c ≥ 6.7% (≥ 50 mmol/mol) should be treated as having probable (overt) diabetes in pregnancy (DIP). At 24 to 28 weeks, those who had an initial HbA1c of ≤5.8% (≤40 mmol/mol) are offered a 50 g GCT, while those with an initial HbA1c 5.9–6.6% (41–49 mmol/mol) are offered a two-hour 75 g OGTT, using the NZSSD GDM criteria. Lower treatment targets were also recommended for the fasting blood glucose level (BGL) as described in Table 1 [10]. A summary of the diagnostic approaches and treatment targets are presented in Table 1. What remains unclear is whether this evolution in screening, diagnostic criteria, and treatment thresholds in New Zealand and Australia have led to improved pregnancy and neonatal outcomes for women with GDM, given the increasing numbers of patients being diagnosed, and the pressure it places on finite resources [11,12].

Further, certain ethnic groups are known to have a higher prevalence of GDM [14]. Between ethnic groups there are known differences in body mass index (BMI) which affect the outcomes of GDM [15]. GDM is also associated with macrosomia, miscarriage, preterm labour, increased Neonatal Intensive Care Unit (NICU) admission, hyperbilirubinaemia requiring phototherapy, and perinatal death [1]. A study in 2000 by Metcalfe showed that in New Zealand, the prevalence of BMI > 30 kg/m^2^ in Pasifika women was 71.7% and Māori women was 41.9%: significantly higher than the 14.6% among European women [15]. Australian obesity data vary with changing ethnic mix over time, although one multiethnic area reported overweight and obesity prevalence between 2008 to 2018 consistently greater than 50% of the population [16]. A recent western Sydney study showed that BMI of women with GDM was on average 25.6 kg/m^2^ in 2016 [17].

This study aims to compare the differences in approach between Australia and New Zealand to diagnose and treat GDM using data from one site in each country. Research questions include (i) were NZ’s introduction of a booking HbA1c to diagnose DIP early, and use of lower fasting glucose treatment targets associated with improved outcomes? (ii) Was the change from Australia’s use of a traditional risk-based GDM screening approach, to the subsequent adoption of the IADPSG diagnostic criteria and lowering of treatment thresholds associated with an improvement in GDM outcomes? (iii) Were there differences in management and/or pregnancy outcomes between the two sites when the same criteria are used? (iv) Were there significant differences in pregnancy outcomes between ethnic groups for women treated for GDM after adjustment for BMI?

## 2. Methodology

Data were collected prospectively for the years 2007 to 2018 on women who presented for obstetric care and were diagnosed with GDM across two tertiary hospitals. Women with pre-existing gestational diabetes, multiple gestations, who gave birth at hospitals other than the two hospitals involved, miscarriages and terminations prior to 20 weeks’ gestation, and women who underwent a diagnostic oral glucose tolerance test (OGTT) at pathology centres outside of the local public laboratories were excluded.

The first site was Waikato Hospital (WH), a tertiary hospital in Hamilton, New Zealand. This is the main hospital of the Waikato District Health Board and services a population of more than 426,300 covering over 21,000 km^2^. It has a distinctly ethnically diverse population, including Māori, European, Asian, and some Pasifika pregnant women. The second site was Liverpool Hospital (LH), an 800-bed tertiary referral centre in south-west Sydney, New South Wales, Australia. LH manages more than 3000 deliveries per year and is a high-risk pregnancy referral centre with the district level 6 nursery facilities.

To diagnose GDM, WH used two screening criteria throughout the period, consistent with the New Zealand guidelines of the day and as described above and detailed in Table 1. The initial group (NZSSD) applies to pregnancies between 2008 and 2015. The second group (NZMOH) and applies to pregnancies between 2016 and 2018.

LH also used two separate GDM diagnostic criteria during the period, with the diagnostic criteria used between 2007 to January 2016 based on the 1998 Australian Diabetes in Pregnancy Society (ADIPS98) as shown in Table 1 [8]. In February 2016, LH adopted the updated ADIPS/IADPSG guidelines [9,18]. Table 1 also shows the change in, treatment targets for both hospitals. WH’s initial treatment target were fasting glucose ≤ 5.5 mmol/L and/or 2-h post-prandial glucose ≤ 6.7 mmol/L. This changed during the period, in line with the NZMOH’s guidelines [10]. Between 2008 to 2016, LH’s target fasting glucose was ≤5.5 mmol/L and/or 2-h post prandial glucose ≤ 7.0 mmol/L, in line with the earlier ADIPS [8]. After adoption of the updated ADIPS 2014 guidelines, their treatment targets reduced to fasting glucose ≤ 5.3 mmol/L, 1 h post-prandial glucose ≤ 7.4 mmol/L, and/or 2-h post prandial glucose ≤ 7.0 mmol/L.

A composite of adverse neonatal outcomes was used to reflect adverse effects from exposure to maternal hyperglycaemia which is possibly modifiable with identification of hyperglycaemia in pregnancy and maternal treatment. The composite included macrosomia, perinatal death, preterm delivery (before 37 weeks), neonatal hypoglycaemia, and phototherapy for jaundice. This was modified from the Metformin in Gestational diabetes Trial which used a 6 factor composite [19]. Macrosomia was defined as >4500 g for purposes of this study in line with Boulet et al. [20], but also as Māori and Pasifika populations tend to have higher birthweight babies, with up to 25% having birthweight of >4000 g as reported by Rao et al. [21]. Low birth weight was defined as <2500 g [22].

The main analysis compares the four groups defined in Table 1. Adjustments for confounders of age, parity, body mass index, ethnicity, past history of gestational diabetes, family history of diabetes, and smoking history were performed for continuous outcome variables.

A sub-analysis was also conducted where we further defined 2 groups of women who had OGTT results consistent with the NZSSD/NZMOH diagnostic criteria of fasting level ≥5.5 mmol/L and/or 2-h level ≥9.0 mmol/L across both WH and LH to determine whether there were significant differences in this cohort for adverse outcomes, adjusted for confounders.

We also simplified ethnicity into 5 groupings: Anglo-European, South-East Asian/East Asian, South Asian and Middle Eastern, Māori and Pasifika (grouped due to smaller number at LH), and others. A further sub-analysis compared their GDM outcomes by ethnic group, adjusted for BMI.

### 2.1. Ethics

Ethics approval was obtained from the Western Sydney University Human Research Ethics Committee (HREC Approval Number: H13648).

### 2.2. Statistics

Deidentified data were collated and analysed using Microsoft Excel. Statistical analysis was performed using IBM SPSS Statistics for Windows, Version 27.0. Armonk, NY: IBM Corp.

Categorical variables were described using numbers and percentages and continuous variables using mean (95% confidence intervals) or median and range. Chi-square test and one-way analysis of covariance was performed to compare between groups and provide significance values at 95% confidence intervals. Binomial logistic regression and one-way analysis of covariance were performed to adjust for confounders. All tests are 2 tailed, with *p* < 0.05 taken as significant.

## 3. Results

Overall, 7518 singleton pregnancy records were included. Table 2 shows the characteristics of the 2278 and 5240 women at WH and LH, respectively, including the numbers before and after the screening, diagnostic, and treatment changes.

Table 3 details maternal treatment and outcomes. Continuous variables were adjusted for age, parity, body mass index, ethnicity, past history of gestational diabetes, family history of diabetes, and smoking history; crude values unadjusted for the listed factors were also presented. Table 4 looked at results of the odds ratio of the composite outcomes compared to women of Anglo-European ethnicity from the Liverpool IADPSG group. For our cohort of women we performed a logistic regression of low birth weight births and found it was significantly negatively associated with BMI (OR 0.94, 95% CI: 0.92–0.97), hypertension (OR 0.45, 95% CI: 0.23–0.85) and pre-eclampsia (OR 0.22, 95% CI: 0.10–0.49) during pregnancy; it was not significantly associated with HbA1c, either on diagnosis or in the third trimester, ethnicity, BMI, age, parity, family history of diabetes, or smoking history.

Table 5 analysed the odds ratios of ethnic differences on maternal and foetal outcomes comparing with the Anglo-European ethnic group, adjusted for BMI. Table 6 shows the adjusted outcomes between the WH and LH cohorts with a positive OGTT according to the NZSSD/NZMOH criteria.

## 4. Discussion

We have shown that the adjusted composite outcome, for the clinically important complications at birth, was lowest in the IADPSG cohort, refer Table 4. We have also shown that the introduction of a booking HbA1c and use of lower fasting glucose treatment targets were not associated with improved outcomes, that while there were differences in management and some individual outcomes between WH and LH, the composite outcomes were similar and after adjusting for BMI there were no significant differences in pregnancy outcomes between ethnic groups.

There have been previous concerns that New Zealand’s approach, which differs from the IADPSG guidelines, could have an adverse impact on GDM outcomes [3,23]. The diagnostic HbA1c was lower for the IADPSG group compared to the other groups and also the 3rd trimester HbA1c showed improvement for both sites with the newer diagnostic criteria. The diagnostic OGTT results showed that WH had a paradoxically lower average fasting level (5.04 and 5.02 mmol/L compared to 5.25 and 5.15 mmol/L) but much higher average 2-h glucose levels than LH (9.2 and 9.38 mmol/L compared to 8.25 and 7.52 mmol/L). The NZSSD and NZMOH approach overall uses higher OGTT cut offs than the IADPSG which diagnoses women with fasting levels ≥5.5 mmol/L, and higher 2-h cut-offs at ≥9.0 mmol/L.

There is a trial underway looking to determine whether using the lower IADPSG diagnostic OGTT cut-offs compared to using the higher cut-offs in the same ranges as the NZSSD/NZMOH will reduce maternal and perinatal morbidity [24]. A systematic review [25] and a study in Western Australia [26] showed that adoption of the IADPSG has resulted in 20 to 75% increase in the diagnosis of GDM, however both the Western Australia study and a 2017 review concluded that this was justified due to improvements in perinatal morbidity, and potential long-term benefits [26,27]. More recently Hillier et al. randomised 23,792 women to perform either the one step approach with diagnostic cut-off similar to the IADPSG or a two-step approach with 50 g GCT then a 100 g OGTT and concluded there were no significant between-group differences in perinatal and maternal outcomes. It is arguable that the study was sufficiently powered to detect true differences due to the low rate of reported perinatal and maternal outcomes [28]. Reviews on the use of a one-step two-hour OGTT have commented there is increased convenience but higher costs compared to the two-step approach with an initial GCT or a fasting glucose for diagnosis [29,30]. There is a trial underway involving approximately 65,000 women in Sweden, analysing the changeover in criteria from older Swedish criteria with varying cut-offs to the IADPSG criteria in a stepped wedge cluster randomised controlled trial [31].

There were several local changes to local policies over time that could explain some of the differences between groups. HbA1c was not routinely tested in both sites for diagnosis or in the 3rd trimester. WH initially routinely tested HbA1c levels in the 3rd trimester, but only adopted early HbA1c testing with the NZMOH criteria. LH stopped routinely testing for HbA1c levels in the 3rd trimester and hence the 3rd trimester HbA1c was not a reliable marker of their treatment outcomes. Furthermore, in LH there was an increase in the use of metformin, particularly after the favourable results from the MIG study [19], which likely cause a lowering of the short-acting insulin use in the IADPSG group. Additionally, the local policy surrounding NICU admission likely shows variation over time, with higher numbers of GDM neonates being admitted in the latter groups for both sites.

Table 6 details a comparison of the adjusted treatment outcomes between WH and LH using the higher OGTT cut-offs of the NZSSD/NZMOH GDM criteria. This more or less echoed the earlier findings of the maternal and foetal outcomes in Table 3. Surprisingly, despite WH having lower treatment targets than LH, there was no difference between WH and LH in composite outcome, with LH even having some favourable outcomes. Furthermore, this GDM subset represents a higher risk group, with an increased adverse composite outcome rate compared to the overall IADPSG cohort. This echoes similar findings from other studies who have looked at countries who have introduced the IADPSG OGTT diagnostic cut-offs and found improved maternal and perinatal outcomes compared to groups using higher diagnostic cut-offs [26,32,33,34,35].

The addition of the HbA1c as part of the algorithm to screen for pre-existing diabetes or early GDM in pregnancy has been identified as a useful tool by various studies, with its high utility and convenience [36,37,38,39]. Some studies [18,40] and guidelines [41] have argued against using HbA1c in the third trimester as an indicator of glycaemic control during pregnancy. A recent paper from the Vitamin D and Lifestyle Intervention for GDM prevention cohort showed that use of an HbA1c threshold of ≥5.7% (39 mmol/mol) before 20 weeks had low sensitivity for predicting GDM or adverse outcomes in overweight/obese European women [42]. WH using the NZMOH criteria and adding in a screening HbA1c, effectively removed the higher risk DIP cohort from our comparison [43]. This was not reflected in improved outcomes among the remaining women diagnosed with GDM in NZMOH, although whether there were improved outcomes for the women diagnosed with DIP was outside the scope of this study.

In reality, the WH cohorts started off with higher glucose values on average and they also had a shorter period of intervention. The LH cohorts were diagnosed and presumably commenced on treatment on average 4 weeks earlier than WH GDM patients, which may play a large role in reducing adverse outcomes.

What the optimal treatment targets are, and whether the benefits of a lower treatment target outweighs the costs, are contentious issues [44]. New Zealand’s treatment targets have become the most stringent over time [45] with the NZMOH criteria lowering the fasting glucose target from ≤5.5 mmol/L to ≤5.0 mmol/L. In comparison LH have lowered both fasting targets (from ≤5.5 to ≤5.3 mmol/L) but kept the 2-h treatment targets unchanged at ≤7.0mmol/L). The WH cohorts had overall a lower perinatal death rate, but otherwise the IADPSG cohort had a much-improved composite outcome compared to even the NZMOH cohort with the lowest treatment targets. This seems to suggest that glucose target levels alone are not the main cause of the improvement in outcomes. Putting this together with results in Table 5, the impact of New Zealand’s high obesity rates on maternal and perinatal outcomes merits consideration [46,47,48]. Studies on Fijian women affected by GDM, a Pacific Island nation who have higher average BMIs than European women, found worsened outcomes for those in the higher obesity categories [49,50].

Our study showed no ethnic differences in composite outcomes after adjusting for BMI, although some individual outcomes did differ between Anglo-European and other ethnic groups. This suggests that maternal BMI is an independent factor for adverse pregnancy outcomes. Many international studies have reported that obesity is an independent risk factor for poor maternal outcomes, and have focused on the pre-conception weight as a target for intervention [51,52,53]. There are also studies that suggest women of ethnicities other than Anglo-European tend to have more than recommended weight gain during pregnancy, and worse outcomes [54,55,56].

A strength of the study is the inclusion of large datasets prospectively collected over 12 years across two sites and the opportunity to compare changes in gestational diabetes diagnoses and management. This however results in a weakness for this study that there were differences in data coding as new data fields were added to over time resulting in some inconsistencies in the amount of data collected for the earlier years. Comparisons across different sites in the future would be best served with use of a uniformly adopted dataset such as the ADIPS dataset [57]. Some of the data were collected from obstetric records, particularly for postnatal progress, and there could be discrepancies in the way these were coded and interpreted.

Limitations of this in study included that we categorised ethnicities into 5 groups for ethnic differences, and we had to create broad ethnic groups such as Māori and Pacific Islands, South Asian and Middle East, and Other, to enable comparison and adjust across 2 sites. In relation to Anglo-European ancestry, this may differ greatly across WH and LH, and may not be representative of Anglo-European ancestry generally in other parts of the world [58]. Similarly, Māori and Pacific Islands people have been shown to differ in some pregnancy outcomes [42].

Only one hospital was represented in each country, hence the results are not necessarily representative of either country. There was no measurement of socio-economic status, which can vary greatly by geographical locations. Patient factors such as adherence with treatment and achievement of glucose targets are hard to quantify and can particularly confound the results.

## 5. Conclusions

Our study suggests that the use of the IADPSG guidelines led to more favourable perinatal outcomes among women diagnosed with GDM when compared with those diagnosed using New Zealand and the ADIPS 1998 recommendations. It is unclear if this is due to the inclusion of lower risk women following the criteria changes. The lack of change in outcomes after lowering of the targets in New Zealand suggests that their use should be reviewed. The importance of BMI in the ethnic differences in the composite outcome suggest that more research into pre-conception and first trimester lifestyle interventions is needed to safely improve pregnancy outcomes among these women.

## Figures and Tables

**Table 1 ijerph-18-04588-t001:** Comparison of Liverpool and Waikato Diagnostic and Treatment Approaches for Gestational Diabetes Diagnosis.

	Waikato NZSSD 1995 [10]	Waikato NZMOH 2014	Liverpool ADIPS 1991 [7,8]	Liverpool IADPSG [9,13]
SCREENING APPROACH				
	Universal GCT	Booking HbA1c (<20 weeks) If ≤40 mmol/mol (5.8%), do GCT if 41–49 mmol/mol (5.9–6.6%), do OGTT ≥50 mmol/mol (6.7%), referral to diabetes in pregnancy service	Risk-based screening: Low risk groups: GCT at 26–28 weeks, if ≥7.8 mmol/L then OGTT High risk groups *: OGTT at time of antenatal booking, if normal repeat OGTT at 26–28 weeks	Universal OGTT at 24–28 weeks Early OGTT performed at any time for those with 1 high risk factors or 2 moderate risk factors *
RISK FACTORS FOR SCREENING				
			Risk factors include glycosuria, age > 30 years, obesity, family history of diabetes, past history of GDM or glucose intolerance, previous adverse pregnancy outcome, belonging to an ethnic group with a high risk for GDM (e.g., Indigenous, Polynesian, Middle Eastern and Asian).	High risk factors include: previous GDM, elevated BGL, age ≥ 40 years, 1st degree family history (or sister with GDM), BMI > 35 kg/m^2^, previous macrosomia (>4500 g or >90th centile), polycystic ovarian syndrome, on corticosteroids or antipsychotics. Moderate risk factors include: Asian, Indian subcontinent, Aboriginal, Torres Strait Islander, Pacific Islander, Maori, Middle Eastern, or non-white African ethnicity. BMI 25–35 kg/m^2^.
ORAL GLUCOSE TOLERANCE TEST DIAGNOSTIC CUT OFFS				
1-h 50 g GCT cut offs:	- ≥11.1 mmol/- referral to diabetes in pregnancy service - 7.8–11.0 mmol/L, proceed to OGTT	If HbA1c at booking is ≤40 mmol/mol (5.8%) If ≥7.8 mmol/L, proceed to OGTT		
OGTT fasting level (mmol/L)	≥5.5	≥5.5	≥5.5	≥5.1
OGTT 1-h level				≥10.0
- OGTT 2-h level (mmol/L)	≥9.0	≥9.0	≥8.0	≥8.5
TREATMENT TARGETS				
- Fasting BGL (mmol/L)	≤5.5	≤5.0	≤5.5	≤5.3
- 1-h BGL (mmol/L)		≤7.4	N/A	≤7.4
- 2-h BGL (mmol/L)	≤6.5	≤6.7	≤7.0	≤7.0

BGL = blood glucose level, GCT = glucose challenge test, ADIPS = Australian Diabetes in Pregnancy Society, IADPSG = International Association of Diabetes and Pregnancy Study Groups, NZMOH = New Zealand Ministry of Health, NZSSD = New Zealand Society for Study of Diabetes, OGTT: oral glucose tolerance test. * Risk factors described in section “Risk Factors for Screening”

**Table 2 ijerph-18-04588-t002:** Demographic Data.

	Waikato NZSSD (*n* = 1344)	Waikato NZ MOH (*n* = 934)	Liverpool ADIPS 1991 (*n* = 3452)	Liverpool IADPSG (*n* = 1788)	*p*-Value Across 4 Groups
Age (95% CI)	31.4 (31.1–31.7)	31.6 (31.3–32.0)	31.4 (31.2–31.5)	31.0 (30.7–1.2)	0.016
Parity (95% CI)	1.3 (1.2–1.4)	1.1 (1.0–1.2)	1.3 (1.3–1.4)	1.3 (1.2–1.3)	0.006
BMI (kg/m^2^) (95% CI)	33.3 (32.8– 33.7)	32.1 (31.6–32.6)	27.5 (27.3–27.8)	28.3 (27.9–28.6)	<0.001
ETHNICITY					<0.001
- Anglo European	612 (45.6%)	342 (36.6%)	779 (25.8%)	397 (22.3%)	
- SE/E Asian	156 (11.6%)	154 (16.5%)	774 (22.4%)	332 (18.6%)	
- South Asian and ME	126 (9.4%)	172 (18.4%)	1483 (43.0%)	809 (45.2%)	
- Māori and PI	325 (24.2%)	222 (23.8%)	201 (5.8%)	139(7.8%)	
- Other	66 (4.9%)	25 (2.7%)	152 (4.4%)	103 (5.8%)	
- Unreported	59 (4.4%)	19 (2.0%)	63 (1.8%)	2 (0.1%)	
GDM History	228 (17.0%)	175 (18.7%)	780 (22.6%)	394 (22.1%)	<0.001
Family History	623 (46.4%)	376 (40.3%)	1627 (47.1%)	784 (43.8%)	0.001
Smoking during pregnancy	162 (12.1%)	89 (9.5%)	207 (6.0%)	110 (6.2%)	<0.001

BMI = body mass index; GDM = gestational diabetes, SE/E = South East/East, ME = Middle East, PI = Pacific Islands, NZSSD = New Zealand Society for Study of Diabetes, NZMOH = New Zealand Ministry of Health, ADIPS = Australian Diabetes in Pregnancy Society, IADPSG = International Association of Diabetes and Pregnancy Study Groups. 95% CI = 95% confidence interval.

**Table 3 ijerph-18-04588-t003:** Maternal and Foetal Outcomes.

	Waikato NZSSD (*n* = 1344)	95% CI	Waikato NZMOH (*n* = 934)	95% CI	Liverpool ADIPS 1998 (*n* = 3452)	95% CI	Liverpool Post-IADPSG (*n* = 1788)	95% CI	*p*-Value Across 4 Groups
MATERNAL OUTCOMES									
OGTT result on diagnosis **									
- Fasting glucose (mmol/L)	5.04–adjusted 5.20–crude	4.99–5.09	5.02–adjusted 5.13–crude	4.97–5.08	5.25–adjusted 5.20–crude	5.22–5.28	5.15–adjusted 5.12–crude	5.11–5.19	<0.001
- 1 h glucose (mmol/L)					10.09–adjusted 10.15–crude	9.95–10.23	9.66–adjusted 9.64–crude	9.57–9.74	<0.001
- 2 h glucose (mmol/L)	9.20–adjusted 9.01–crude	9.09–9.32	9.38–adjusted 9.28–crude	9.25–9.51	8.35–adjusted 8.42–crude	8.28–8.42	7.52–adjusted 7.55–crude	7.43–7.61	<0.001
OGTT test week **	27.7–adjusted 27.7–crude	27.55–28.1	27.5–adjusted 27.4–crude	27.03–27.94	23.4–adjusted 23.6–crude	23.17–23.63	22.9–adjusted 23.0–crude	22.61–23.2	<0.001
HbA1 c at diagnosis **	5.6% or 38 mmol/mol–adjusted 5.6% or 38 mmol/mol–crude	5.55–5.62	5.3% or 34 mmol/mol–adjusted 5.4% or 36 mmol/mol–crude	5.33–5.41	5.3% or 34 mmol/mol–adjusted 5.3% or 34 mmol/mol–crude	5.31–5.35	5.2% or 33 mmol/mol–adjusted 5.2% or 33 mmol/mol–crude	5.19–5.25	<0.001
HbA1c during 3rd trimester ** (%)	5.6% or 38 mmol/mol–adjusted 5.6% or 38 mmol/mol–crude	5.51–5.58	5.5% or 37 mmol/mol–adjusted 5.5% or 37 mmol/mol–crude	5.42–5.51	5.5% or 37 mmol/mol–adjusted 5.2% or 33 mmol/mol–crude	5.49–5.55	5.4% or 36 mmol/mol–adjusted 5.4% or 36 mmol/mol–crud	5.31–5.47	0.001
Medical nutritional therapy alone	684 (50.9%)		462 (49.5%)		1874 (54.3%)		966 (54.0%)		0.017
Metformin treatment	164 (12.2%)		113 (12.1%)		128 (4.1%)		279 (15.9%)		<0.001
Short-acting insulin treatment	485 (36.1%)		312 (33.4%)		1146 (33.2%)		363 (20.3%)		<0.001
Long-acting insulin treatment	407 (30.3%)		293 (31.4%)		1150 (33.4%)		554 (31.6%)		0.087
Hypertension in pregnancy	52 (3.9%)		33 (3.5%)		184 (6.1%)		82 (4.6%)		<0.001
PET	33 (2.5%)		16 (1.7%)		37 (1.1%)		37 (2.1%)		0.002
NEONATAL OUTCOMES									
Delivery weeks **	37.9–adjusted 37.9–crude	37.82–38.02	38.3–adjusted 38.3–crude	38.21–38.45	38.7–adjusted 38.7–crude	38.62–38.75	38.6–adjusted 38.6–crude	38.51–38.68	<0.001
Preterm delivery (<37 weeks)	139 (10.3%)		75 (8.0%)		276 (8%)		164 (9.2%)		0.051
Birth weight ** (g)	3359.4–adjusted 3418.5–crude	3325.8–3392.9	3387.6–adjusted 3410.8–crude	3349.3–3425.8	3313.1–adjusted 3285.0–crude	3292.3–3334.0	3329.2–adjusted 3331.7–crude	3302.5–3355.9	0.005
Macrosomia (>4500 g)	47 (3.5%)		15 (1.6%)		52 (1.5%)		18 (1%)		<0.001
Low Birth Weight (<2500 g)	81 (6.0%)		45 (4.8%)		219 (6.3%)		108 (6.0%)		0.388
Male Gender	679 (50.5%)		504 (54%)		1740 (50.6%)		916 (51.3%)		0.308
Spontaneous vaginal delivery	774 (57.6%)		532 (57%)		1524 (43.9%)		665 (37.0%)		<0.001
Assisted delivery	115 (8.6%)		109 (11.8%)		967 (28.0%)		571 (31.9%)		<0.001
Caesarean delivery–emergency indication	253 (18.8%)		162 (17.3%)		363 (10.5%)		207 (11.6%)		<0.001
Caesarean delivery–elective indication	202 (15.0%)		131 (14.0%)		598 (17.2%)		345 (19.3%)		<0.001
Neonatal intensive care admission	218 (16.2%)		267 (28.6%)		205 (5.9%)		286 (16.0%)		<0.001
Neonatal hypoglycaemia	383 (28.5%)		277 (29.7%)		956 (27.7%)		240 (13.4%)		<0.001
Major congenital abnormalities	19 (1.4%)		16 (1.7%)		28 (0.8%)		14 (0.8%)		0.030
Minor congenital abnormalities	31 (2.3%)		7 (0.7%)		131 (3.8%)		33 (1.8%)		<0.001
Phototherapy treatment for neonatal jaundice	60 (4.5%)		55 (5.9%)		30 (0.9%)		2 (0.1%)		0.040
Perinatal death	2 (0.1%)		3 (0.3%)		34 (1.0%)		10 (0.6%)		0.005
Composite Outcome #	510 (37.9%)		349 (37.4%)		1152 (33.4%)		349 (19.5%)		<0.001

ADIPS = Australian Diabetes in Pregnancy Society, IADPSG = International Association of Diabetes and Pregnancy Study Groups, NZMOH = New Zealand Ministry of Health, NZSSD = New Zealand Society for Study of Diabetes, OGTT = oral glucose tolerance test, PET = pre-eclampsia toxaemia, 95% CI = 95% confidence interval ** Adjusted for age, parity, body mass index, ethnicity, past history of gestational diabetes, family history of diabetes, and smoking history. Crude refers to values unadjusted for the above. # Composite outcome consists of pregnancies affected by macrosomia, perinatal death, preterm delivery (before 37 weeks), neonatal hypoglycaemia, and phototherapy for jaundice.

**Table 4 ijerph-18-04588-t004:** Logistic Regression of Composite Outcome #.

	Odds Ratio of Composite Outcome with 95% Confidence Intervals
Liverpool IADPSG	1.000
Waikato NZSSD	2.187 (1.831–2.612)
Waikato NZMOH	2.222 (1.841–2.681)
Liverpool ADIPS 1998	1.910 (1.656–2.203)
Ethnicity	
Anglo-European	1.000
South East Asian and East Asian	0.907 (0.763–1.078)
South Asian and Middle Eastern	1.017 (0.879–1.175)
Māori and Pacific Islands	1.023 (0.935–1.349)
Other	1.042 (0.801–1.355)
Age per year	1.012 (1.001–1.023)
Parity per birth	0.984 (0.942–1.027)
BMI per 1 kg/m^2^	1.022 (1.014–1.030)
History of GDM ^	0.798 (0.699–0.912)
Family history of GDM ^	0.945 (0.849–1.053)
Smoking history ^	0.931 (0.764–1.134)

ADIPS = Australian Diabetes in Pregnancy Society, BMI = body mass index, IADPSG = International Association of Diabetes and Pregnancy Study Groups, NZMOH = New Zealand Ministry of Health, NZSSD = New Zealand Society for Study of Diabetes. # Composite outcome consists of pregnancies affected by macrosomia, perinatal death, preterm delivery (before 37 weeks), neonatal hypoglycaemia, and phototherapy for jaundice. ^ value of 1 indicates yes.

**Table 5 ijerph-18-04588-t005:** Logistic Regression (with 95% Confidence Intervals) by Ethnicity Compared with Anglo European Ethnicity, Adjusted for BMI.

	Anglo-European	SOUTH EAST ASIAN and EAST ASIAN (*n* = 1397)	SOUTH ASIAN and MIDDLE EASTERN (*n* = 2573)	MĀORI and PASIFIKA (*n* = 840)	OTHER (*n* = 340)
Metformin Use	1	1.328 (0.939–1.633)	1.283 (1.033–1.593)	1.457 (1.131–1.593)	1.466 (1.007–2.135)
Short-Acting Insulin Use	1	0.874 (0.743–1.029)	1.140 (1.003–1.296)	0.861 (0.808–1.142)	0.930 (0.723–1.195)
Long-Acting Insulin Use	1	0.729 (0.614–0.865)	1.260 (1.108–1.433)	0.903 (0.759–1.073)	0.845 (0.655–1.090)
Emergency Caesarean Section	1	1.118 (0.897–1.393)	0.960 (0.800–1.151)	1.218 (0.973–1.525)	1.603 (1.186–2.168)
Elective Caesarean Section	1	0.900 (0.734–1.103)	1.201 (1.029–1.402)	0.560 (0.443–0.708)	1.160 (0.868–1.550)
Spontaneous Normal Vaginal Delivery	1	1.223 (1.057–1.416)	0.743 (0.659–0.839)	1.641 (1.391–1.936)	0.659 (0.518–0.837)
Perinatal Death	1	0.969 (0.354–2.652)	1.792 (0.856–3.749)	0.666 (0.183–2.418)	0.557 (0.072–4.330)
NICU Admission	1	0.964 (0.773–1.201)	0.850 (0.709–1.018)	1.280 (1.028–1.592)	1.005 (0.719–1.406)
Neonatal Hypoglycaemia	1	0.832 (0.698–0.991)	1.016 (0.885–1.165)	1.071 (0.893–1.283)	0.965 (0.740–1.260)
Congenital Major Abnormalities	1	1.523 (0.742–3.126)	1.124 (0.598–2.113)	1.193 (0.529–2.694)	1.034 (0.302–3.532)
Congenital Minor Abnormalities	1	1.055 (0.707–1.574)	0.888 (0.622–1.267)	0.715 (0.42–1.217)	1.133 (0.590–2.177)
Composite Outcomes	1	0.852 (0.726–1.0)	0.922 (0.811–1.048)	1.086 (0.917–1.287)	1.006 (0.787–1.285)

**Table 6 ijerph-18-04588-t006:** Comparison of outcomes between women in WH and LH with GDM that returned levels of ≥5.5 mmol/L and/or ≥9.0 mmol/L on their oral glucose tolerance test, consistent with the NZSSD and/or NZMOH criteria.

	Waikato GDM with Positive OGTT per NZSSD/NZMOH (*n* = 2278)	Liverpool GDM with Positive OGTT per NZSSD (*n* = 2403)	Odds Ratio (with 95% CI) Comparing LH Positive OGTT to Waikato NZSSD/NZMOH ^Δ^** (Odds Ratio = 1)	*p*-Value
MATERNAL OUTCOMES				
HbA1c at diagnosis **^α^(%)	5.52 (95% CI: 5.50–5.55)	5.36 (95% CI: 5.33–5.39)		<0.001
HbA1c during 3rd trimester **^α^(%)	5.53 (95% CI: 5.49–5.58)	5.33 (95% CI: 5.27–5.38)		<0.001
Metformin treatment	277 (12.2%)	156 (6.5%)	0.63 (0.49–0.79)	<0.001
Short-acting insulin treatment	797 (35.0%)	870 (36.2%)	1.02 (0.88–1.17)	0.810
Long-acting insulin treatment	700 (30.7%)	998 (41.5%)	2.07 (1.78–2.4)	<0.001
Hypertension in pregnancy	85 (3.7%)	143 (6.0%)	3.21 (2.2–4.4)	<0.001
PET	49 (2.2%)	39 (1.6%)	1.12 (0.69–1.81)	0.657
NEONATAL OUTCOMES				
Delivery weeks **^α^	38.09 (95% CI: 38.01–38.17)	38.59 (95% CI: 38.52–38.66)		<0.001
Preterm delivery (<37 weeks)	214 (9.4%)	211 (8.8%)	1.07 (0.85–1.35)	0.571
Birth weight **^α^ (g)	3390.2 (95% CI: 3363.7–3416.7)	3338.0 (3313.4–3362.7)		0.006
Macrosomia (>4500g)	62 (2.7%)	42 (1.7%)	1.24 (0.79–1.95)	0.349
Male Gender	1183 (51.9%)	1237 (51.5%)	1.00 (0.87–1.14)	0.937
Normal vaginal delivery	1306 (57.3%)	971 (40.4%)	0.43 (0.37–0.49)	<0.001
Caesarean delivery–emergency indication	415 (18.2%)	269 (11.2%)	0.63 (0.52–0.77)	<0.001
Caesarean delivery–elective indication	333 (14.6%)	434 (18.1%)	1.29 (1.07–1.56)	0.007
Neonatal intensive care admission	485 (21.3%)	205 (8.5%)	0.35 (0.29–0.43)	<0.001
Neonatal hypoglycaemia	660 (29.0%)	658 (27.4%)	1.01 (0.87–1.18)	0.871
Major congenital abnormalities	35 (1.5%)	21 (0.9%)	0.47 (0.25–0.90)	0.022
Minor congenital abnormalities	38 (1.7%)	81 (3.4%)	2.49 (1.60–3.90)	<0.001
Phototherapy treatment for neonatal jaundice	115 (5.0%)	19 (0.8%)	0.18 (0.11–0.30)	<0.001
Perinatal death	5 (0.2%)	24 (1.0%)	3.74 (1.30–10.82)	0.015
Composite Outcome #	859 (37.7%)	814 (33.9%)	0.98 (0.85–1.12)	0.736

ADIPS = Australian Diabetes in Pregnancy Society, BMI = body mass index, IADPSG = International Association of Diabetes and Pregnancy Study Groups, NZMOH = New Zealand Ministry of Health, NZSSD = New Zealand Society for Study of Diabetes, 95% CI = 95% confidence interval. ** Adjusted for age, parity, body mass index, ethnicity, gestational diabetes, family history of diabetes and smoking history. ^α^ Averages and 95% confidence intervals were calculated using ANCOVA for continuous variables. ^Δ^ Odds ratios and 95% confidence intervals are calculated using binary logistic regression for categorical variables. # Composite outcome consists of pregnancies affected by macrosomia, perinatal death, preterm delivery (before 37 weeks), neonatal hypoglycaemia and phototherapy for jaundice.

## Data Availability

The data are not publicly available due to privacy reasons as they were part of audit activities.

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
