# Peer review of "Comparison of Pregnancy Outcomes Using Different Gestational Diabetes Diagnostic Criteria and Treatment Thresholds in Multiethnic Communities between Two Tertiary Centres in Australian and New Zealand: Do They Make a Difference?"

_ijerph, 2021, doi:10.3390/ijerph18094588_

Round 1

Reviewer 1 Report

The paper entitled “Comparison of pregnancy outcomes using different criteria and treatment thresholds in multiethnic communities in Australia and New Zealand: Do they make a difference?” aims to compare pregnancy outcomes using these different diagnostic approaches. This topic is important and interesting. I would recommend the authors to address the following points when revising the manuscript.

1. I noticed that the author name“Vincent W. Wong”appeared twice,please confirm.

2.  According to the manuscript,the study was a prospective study, not a retrospective one. Please confirm and state the reason.

3. From Table 1, the author obtained some results, such as “Women from WH had a significantly higher BMI, a greater proportion of smokers but lower proportion of women with previous GDM than women from LH.”However,in Table 1,“p-value across 4 groups”, the author did not seem to make a pairwise comparison between the groups. Is there a similar situation in Table 3? Please check.

4. After the implementation of IADPSG criteria, there have been some comparative studies. Could you introduce the main conclusions of these comparisons?

5. The convenience of diagnostic methods is an aspect that needs to be considered during diagnosis. Please make a brief comparison and introduction of different methods.

6. Are there any other reports on HbA1c as a screening indicator, and how effective is it?

Author Response

Please see the attachment "Response to Reviewer 1_Yuen.docx"

Reviewer 2 Report

The manuscript "Comparison of pregnancy outcomes using different criteria and treatment thresholds in multiethnic communities in Australia and New Zealand: Do they make a difference?" looks very interesting. A major revision is required to improve the manuscript quality.  

Provide ethical approval details of the study

Is there any correlation between HbA1C level and low birth weight?

What are the inclusion and exclusion criteria for sample selection?

At the end of the discussion, explain what the limitation of the study is.

Author Response

Please see the attachment "Response to Reviewer 2_Yuen.docx"

Reviewer 3 Report

The paper presents data from a large ‘real life’ experiment looking at a composite neonatal outcome using a variety of clinical models with differing GDM screening criteria and glucose targets for pregnancy.

As single sites are not representative of the country as a whole, the title should be revised to reflect that it is a comparison of two tertiary centres.

Comparisons are somewhat problematic due to the number of factors that are different between the cohorts. Would it be more productive to compare LH before and after the change, then present WH before and after the change in criteria?

The description of the outcomes in the results section could be improved as to ensure it adds to the readers understanding and does not repeat data available in the table.

I assume that the sub-analysis (Aus subset positive using NZSSD) was identified to determine whether it was the criteria or other factors driving the differences between groups? As they were not treated to the same targets, this is less helpful and should be removed.

The discussion could include thoughts on the recent publication Hillier, T A Pragmatic, Randomized Clinical Trial of Gestational Diabetes Screening. NEJM 2021.DOI: 10.1056/NEJMoa2026028

Minor points:

Explain what is meant by ‘adjusted HbA1c at diagnosis’

Table 1 has poor readability.

Table 2 not well aligned, so difficult determine, for example, BMI in the Liverpool IADPSG group.

Table 3 should be reformatted; what is the difference between ‘adjusted’ and ‘crude’ delivery weeks and birth weight.

Table 4, consider reformatting so the reference population is at the top of each section..

Table 5 has figures in bold, perhaps to highlight those that do not cross 1.0? Review consistency.

Are there two authors called Vincent W. Wong? It seems that the authors should be Lili Yuen *, Vincent W Wong, Louise Wolmarans, David Simmons?

Please provide a reference for ADIPS dataset.

Consider removing any unnecessary acronyms to improve readability.

Author Response

Please see the attachment "Response to Reviewer 3_Yuen.docx"

Round 2

Reviewer 2 Report

The manuscript "Comparison of pregnancy outcomes using different gestational diabetes diagnostic criteria and treatment thresholds in multi-ethnic communities between two tertiary centres in Australian and New Zealand: Do they make a difference?" now looks good and is acceptable for publication.

Reviewer 3 Report

The corrections have all been addressed in the revised manuscript.